# 3D Printed Fully Recycled TiO_2_-Polystyrene Nanocomposite Photocatalysts for Use against Drug Residues

**DOI:** 10.3390/nano10112144

**Published:** 2020-10-28

**Authors:** Maria Sevastaki, Mirela Petruta Suchea, George Kenanakis

**Affiliations:** 1Institute of Electronic Structure and Laser, Foundation for Research & Technology-Hellas, N. Plastira 100, 70013 Heraklion, Crete, Greece; msevastaki@iesl.forth.gr; 2Department of Chemistry, University of Crete, 71003 Heraklion, Crete, Greece; 3National Institute for Research and Development in Microtechnologies (IMT-Bucharest), 1 26 A, Erou Iancu Nicolae Street, P.O. Box 38-160, 023573 Bucharest, Romania; 4Center of Materials Technology and Photonics, Hellenic Mediterranean University, 71004 Heraklion, Crete, Greece

**Keywords:** metal oxide nano-structures, titanium dioxide (TiO_2_), photocatalysis, paracetamol, 3D printing, fused filament fabrication (FFF), fused deposition modeling (FDM)

## Abstract

In the present work, the use of nanocomposite polymeric filaments based on 100% recycled solid polystyrene everyday products, enriched with TiO_2_ nanoparticles with mass concentrations up to 40% *w/w*, and the production of 3D photocatalytic structures using a typical fused deposition modeling (FDM)-type 3D printer are reported. We provide evidence that the fabricated 3D structures offer promising photocatalytic properties, indicating that the proposed technique is indeed a novel low-cost alternative route for fabricating large-scale photocatalysts, suitable for practical real-life applications.

## 1. Introduction

For many years, human activities have polluted the environment in many ways; several organic residuals originating from highly toxic pollutants such as pharmaceuticals can be found in water, comprising a critical health and environmental issue [1,2,3]. In many cases, unused, expired and residual pharmaceuticals are discharged into the sewerage system, burdening the aquifer. Moreover, these compounds can also be discharged into the environment through the metabolism of human bodies [4,5,6,7,8]. As a result, pharmaceuticals have been found in sewage, surface and ground water in many countries [6,8,9].

The most common methods used to overcome this problem include the return of medicines to pharmacies and not throwing expired medicinal products in the sewer, as well as biological degradation, chlorination or ozonation, but these are not efficient enough to remove these compounds from the treated water [10,11,12]. These drugs residues must be eliminated using an oxidation method, and advanced heterogeneous photocatalysis seems to be one of the most promising approaches, since it implies the use of an inert catalyst, non-hazardous oxidants and ultraviolet (UV) and/or visible light input [13,14,15,16,17,18,19,20,21,22,23,24,25].

Heterogeneous photocatalysis using TiO_2_ is a method generating free radical •OH using atmospheric air instead of O_3_ or H_2_O_2_, significantly reducing processing costs. The process takes place at ambient conditions and leads to the complete decomposition of both liquid and gaseous pollutants [26,27]. The greatest advantage of this method is that an environmentally friendly catalyst which is widely available, inexpensive, non-toxic, and photo-stable with respect to other photocatalysts is readily available and readily regenerable for the purpose of reuse, maintaining equally high performance for a large number of catalytic cycles [13,14,15,16,17,26]. Several studies have shown that titanium dioxide (TiO_2_) is the most potent semiconductor for the oxidative destruction of organic compounds. TiO_2_ has, in addition to its large photocatalytic activity, greater resistance to corrosion and photo-corrosion, resulting in the possibility of recycling [28].

The disadvantage of photocatalysis when the semiconductor is used in the form of a powder is the need to remove it after the end of the treatment [29,30]. For this reason, international efforts are focused on photocatalytic systems, where the catalyst is used in the form of a film on inert substrates to eliminate the stage of powder removal [31,32,33,34,35]. Over and above that, photocatalytic efficiency increases with effective surface area, and consequently a nanostructured photocatalyst is beneficial. However, solid catalysts’ samples, such as thin films or nanostructured ones, in most cases cannot exceed an overall size of a few centimeters, due to the limitation of the fabrication techniques, limiting their potential use in real-life applications.

In the last few years, 3D printing technology become of great interest in several fields of research, such as medicine, chemistry and materials science, as an effective, fancy, quick and low-cost route for the production of 3D large-size samples [36,37,38,39,40]. The most common technique is fused deposition modeling (FDM) in which polymers are the usual materials used as filaments. It should be noted that although there are several reports on 3D structures for novel environmental applications [41,42,43], there are only a few in which custom-made filaments (with nanoparticles of inorganic materials into a polymeric matrix) are used in combination with FDM technology, i.e., in [44,45], and none for drug-residuals’ removal by photocatalysis.

This work discusses an investigation of the photocatalytic degradation of paracetamol (also known as acetaminophen, APAP), a medicine available in a huge number of countries worldwide, used to treat pain and fever, using 3D-printed photocatalysts enriched with 20% *w/w* nanostructured TiO_2_. It is worth mentioning that the polymeric filaments used to produce these 3D-printed photocatalysts are based on 100% recycled solid polystyrene (PS) everyday products, such as containers, lids, CD cases etc., following an eco-friendly environmental approach. APAP was chosen for this study due to its high occurrence as a pharmaceutical pollutant in environment. As shown in many studies, it was found worldwide in almost all kinds of water source as well in the soil [46]. APAP is reported to be one of the most frequently detected pharmaceuticals in sewage treatment plant effluents [47]. The increasing concentrations of APAP together with other emerging contaminants result in the occurrence of toxic phenomena in non-target species present in receiving aquatic environments. An excellent review regarding the toxic effects of environmental APAP is presented in Ref. [48].

The present experimental results provided strong evidence that the proposed fully recycled 3D printed photocatalysts are good candidates against the degradation of APAP drug residues, reaching an efficiency of almost 75% of a 100 ppm APAP aqueous solution under UV irradiation for 20 min, and ~60% after three cycles of reuse in 200 ppm APAP aqueous solutions, respectively.

## 2. Experimental Details

### 2.1. Synthesis of the Metal Oxide Polymeric Nanocomposites

First, several everyday PS products, such as containers, lids, CD cases etc. were ground using an IKA A11 Basic Analytical Mill (IKA-Werke GmbH and Co. KG, Staufen, Germany) equipped with a high-grade stainless-steel beater, coated with chromium carbide. Recycled grinded PS powder/beads of ~0.2 mm diameter were dissolved in toluene (Merck KGaA, Darmstadt, Germany) (in a sealed bottle, under continuous stirring for 2 h) to create a 20% *w/w* solution. The resultant solution was stirred for 24 h at room temperature using a magnetic stirrer to yield a homogeneous, milky solution.

Subsequently, 2 g and 4 g of commercially available TiO_2_ nanoparticles (TiO_2_ P25 with a mean particle size of ~25 nm, obtained from Evonic Industries AG, Essen, Germany) were introduced in 10 mL of the PS/toluene solution mentioned above under stirring at 40 °C for 30 min, in order to obtain the TiO_2_ homogeneous suspensions with 20% *w/w* and 40% *w/w* concentration in PS, respectively.

Each of the TiO_2_ homogeneous suspensions was transferred to 200 mL of ethanol (95% purity; Merck KGaA, Darmstadt, Germany), to form a dense precipitate, which consisted of the PS and the suspended metal oxide nanoparticles. After formation, the precipitate was collected and dried at 60 °C for 24 h using a typical laboratory oven (Memmert UNP 500 Memmert GmbH + Co., Schwabach, Germany). Using the procedure above, 20 g of 20% *w/w* TiO_2_/PS, and 20 g 40% *w/w* TiO_2_/PS solid nanocomposite solutions were produced, respectively.

### 2.2. Filament Production

The produced TiO_2_/PS solid nanocomposite solutions were cut in ~2–3 mm^2^ pieces, and forwarded to a “Noztek Pro” (Noztek, Shoreham, West Sussex, UK) high temperature extruder, and processed at 240 °C, in order to transform them to a cylindrical filament with a diameter of 1.75 ± 0.15 mm, suitable for 3D-printing. All extrusion parameters, such as extrusion velocity and temperature, were optimized towards the production of a uniform continuous cylindrical cord, with an overall length of ~5 m.

### 2.3. Production of 3D-Printed Photocatalytic Structures

Flat, rectangular-shaped (10 mm × 10 mm × 1 mm) 3D structures were designed using “Tinkercad”, a free online 3D design and 3D printing software from Autodesk Inc (Mill Valley, CA, USA). A dual-extrusion FDM-type 3D printer (Makerbot Replicator 2X; MakerBot Industries, Brooklyn, NY, USA) was used for the direct fabrication of TiO_2_/PS nanocomposite photocatalytic samples, using the PS/TiO_2_ nanocomposite filaments described above. The FDM process of building a solid object involves heating of the fed filament and pushing it out layer-by-layer through a heated (240 °C) nozzle (0.4 mm inner diameter) onto a heated surface (80 °C), via a computer controlled three-axis positioning system (with a spatial resolution of approximately 100 μm in the z-axis and 11 μm in x and y).

### 2.4. Characterization and Photocatalytic Experiments

X-ray diffraction (XRD) measurements were performed in order to determine the crystalline structure of the 3D-printed samples, using a Rigaku RINT 2000 (Rigaku, Tokyo, Japan) diffractometer with Cu Kα (λ = 1.5406Å) X-rays for 2θ = 20.00–60.00° for TiO_2_/PS nanocomposite-based samples and a step time 60°/sec.

Furthermore, Raman spectroscopy measurements were performed at room temperature using a Horiba LabRAM HR Evolution (HORIBA FRANCE SAS, Longjumeau, France) confocal micro-spectrometer, in backscattering geometry (180°), equipped with an air-cooled solid-state laser operating at 532 nm with 100 mW output power. The laser beam was focused on the samples using a 10× Olympus (OLYMPUS corporation, Tokyo, Japan) microscope objective (numerical aperture of 0.25), providing ~14 mW power on each sample. Raman spectra over the 100–700 cm^−1^ wavenumber range (with an exposure time of 5 s and 3 accumulations) were collected by a Peltier cooled CCD (1024 × 256 pixels) detector (HORIBA FRANCE SAS, Longjumeau, France) at −60 °C, with a resolution better than 1 cm^−1^, achieved thanks to an 1800 grooves/mm grating and an 800 mm focal length. Test measurements carried out using different optical configuration, exposure time, beam power and accumulations in order to obtain sufficiently informative spectra using a confocal hole of 100 μm, but ensuring to avoid alteration of the sample, while the high spatial resolution allowed us to carefully verify the sample homogeneity. The wavelength scale was calibrated using a Silicon standard (520.7 cm^−1^) (Silchem Handelsgesellschaft mbH, Freiberg, Germany) and the acquired spectra were compared with scientific published data and reference databases, such as Horiba LabSpec 6 (HORIBA FRANCE SAS, Longjumeau, France).

The photocatalytic activity of the 3D-printed samples was studied by means of the reduction of APAP in aqueous solution, which is a well-known pharmaceutical product that has been used as a model organic to probe the photocatalytic performance of photocatalysts [4,5,7,8]. The investigated samples were placed in a vertical custom-made quartz cell, and the whole setup (cell + solution + sample) was illuminated up to 60 min using an HPK 125 W Philips UV lamp centered at 365 nm (msscientific Chromatographie-Handel GmbH, Berlin, Germany) with a light intensity of ~6.0 mW/cm^2^. The concentration of APAP (degradation) was monitored by UV-Vis spectroscopy in absorption mode (absorption at λ_max_, 665 nm), using a K-MAC SV2100 (K-MAC, Daejeon, Korea) spectrophotometer over the wavelength range of 220–800 nm. In such way, UV-Vis absorption data were collected at 0, 10, 20, 30 and 40 min, while the quantification of the APAP removal (and hence the remaining APAP concentration) was estimated by calculation of the area below the main APAP peak in the range of 220–320 nm. Additional blank experiments (photolysis) without a catalyst were also performed as well as APAP adsorption experiments in the dark.

## 3. Results and Discussion

In order to verify the nominal TiO_2_ loading in PS filaments and 3D-printed nanocomposites, a type of thermogravimetric method was used. A small piece of each sample on a quartz substrate, was weighted and was heated at ~900 °C in order to burn all organics and polymeric residuals, then weighed again. Since TiO_2_ is not affected at all at such temperatures, the remaining mass was the TiO_2_ loading. This way we checked that the nominal TiO_2_ % *w/w* loadings were indeed 20% and 40% *w/w* ± 0.5–1.0%.

Figure 1 depicts a typical optical microscopy photograph of a 3D printed sample (40% *w/w* TiO_2_/PS), as fabricated following the FDM process mentioned above.

As one can notice from Figure 1, rough structures were printed instead of smooth ones, while printing directions were also observed. In our case, the nanoparticle loading (40% *w/w*, shown in Figure 1) in the custom-made filaments, most likely led to low-resolution/low-quality 3D printing. It hence became clear that further investigation was needed in order to improve the printing quality.

Figure 2 presents typical XRD patterns for PS/TiO_2_ 3D printed structures. Well-distinguished diffraction peaks are observed. These correspond to both anatase and rutile phase, in good agreement with the JCPDS card (No. 84–1286) and JCPDS card (No. 88–1175), a crystal structure of anatase and rutile, respectively [49,50] normal for P25 Degussa TiO_2_ that is a mix of the two phases.

Figure 3 shows a typical Raman spectrum of the PS/TiO_2_ 3D printed structures, which exhibit characteristic TiO_2_ phonon frequencies, such as: 143 cm^−1^ (E_g_), 396 cm^−1^ (B_1g_), 516 cm^−1^ (A_1g_) for anatase, and 245 cm^−1^ (two-phonon scattering) and 610 cm^−1^ (A_1g_) for rutile, matching (± 3 cm^−1^) with literature [51,52,53].

The photocatalytic activity of the 3D-printed nanocomposites under UV-A light was evaluated by assessing the reduction of APAP in aqueous solution. The photolytic removal (photolysis) of the pharmaceutical product (in the absence of any photocatalyst) was negligible, underlining the indispensability of the catalysts. Furthermore, to eliminate the possibility of APAP removal by adsorption on the catalysts, the samples were placed at the bottom of the reactor under dark conditions and in contact with the APAP for 30 min, during which time equilibrium of adsorption-desorption was reached. In all cases, removal was insignificant (less than 3%), pointing to the fact that the reduction of the APAP should be attributed to a pure photocatalytic procedure.

The decrease of the concentration of APAP (20 ppm) using both 20% *w/w* and 40% *w/w* 3D-printed TiO_2_/PS nanocomposite samples under UV-A light irradiation is presented in Figure 4. For comparison reasons, the photolysis curve (no catalyst present) is also displayed. According to the photolysis (black curve in Figure 4), the concentration of APAP remained almost constant during ~40 min irradiation, indicating that the photolysis of APAP was almost negligible.

As shown in Figure 4, the 40% *w/w* 3D-printed TiO_2_/PS nanocomposite photocatalysts were highly effective regarding the reduction of APAP compared to the 20% *w/w* 3D printed TiO_2_/PS nanocomposite ones, due to the highly oxidative radicals generated on the TiO_2_ at the surfaces under UV-A irradiation [51].

As already stated, (and shown in the inset of Figure 4), the photodegradation of APAP using the 3D-printed TiO_2_/PS nanocomposite samples followed a first-order kinetics. The calculated apparent rate constants were 0.026 min^−1^ and 0.028 min^−1^ for 20% *w/w* and 40% *w/w* 3D-printed TiO_2_/PS nanocomposite samples, respectively. One can notice that the 40% *w/w* 3D-printed TiO_2_/PS nanocomposite samples are more photocatalytically active than the 20% *w/w* ones, regarding the degradation of APAP, reaching an almost 30% reduction of APAP’s concentration after 10 min of irradiation.

In principle, when a semiconductor is exposed to electromagnetic radiation of an appropriate wavelength, excitation occurs and electrons (e_CB-_) are transferred from the valence band to the conduction band of the material, leaving behind positively charged holes (h_VB+_). The photogenerated holes react with OH^−^ or H_2_O adsorbed on the surface of the catalyst, and hydroxyl radicals that are mainly responsible for the degradation of the target pollutant are created. It is therefore expected that a high recombination rate of photogenerated holes and electrons will be disadvantageous for the performance of the photocatalyst.

Nevertheless, an efficient electron and hole transfer between TiO_2_ depends on the difference between the conduction and valence band potentials of the semiconductor, that should be suitably positioned [54,55]. Concentration of catalysts in the 40% *w/w* 3D-printed TiO_2_/PS nanocomposite is double that in the 20% *w/w* 3D-printed TiO_2_/PS nanocomposite samples thus allowing charge separation and increasing the efficiency of the photocatalytic reaction.

In addition, the apparent rate constant (k) has been calculated as the basic kinetic parameter for the comparison of photocatalytic activities, which was fitted by the equation *ln(C_t_/C*_0_) = −*kt*, where *k* is apparent rate constant, *C_t_* is the concentration of APAP, and *C_0_* is the initial concentration of APAP. It should be noted that the adjusted R-square statistic varies from 0.91499 to 0.94334 indicating that the model used for the determination of the apparent rate constant *(k)* is adequate. The good linear fit of equation *ln(C_t_/C*,_0_) = −*kt* shown in the inset of Figure 4, confirms that the photodegradation for all different concentrations of APAP using 3D printed TiO_2_/PS nanocomposite photocatalysts at 20% and 40% *w/w*, follows first-order kinetics.

It is worth mentioning that the photocatalytic activity tests were carried out at least three times on the 3D-printed TiO_2_/PS nanocomposite samples to examine their stability under UV illumination, demonstrating no changes in the photocatalytic activity after three runs. Moreover, at least three structures with the same TiO_2_ load have been produced and tested, in order to check the reproducibility of the structure manufacturing.

Furthermore, the photocatalytic activity of 40% *w/w* 3D-printed TiO_2_/PS nanocomposite was checked in APAP aqueous solutions with different concentrations (from 20 ppm to 200 ppm). As can be observed from Figure 5a, when the concentration of APAP increases, more irradiation time is needed for its degradation. Although this is expected, it is worth mentioning that for a 100 ppm APAP solution, 20 min of irradiation are enough to reduce it at ~75%, while for 200 ppm APAP solution, 40 min are enough in order to reduce it by the same amount.

Figure 5b confirms that the photodegradation of APAP using 3D printed TiO_2_/PS nanocomposite samples follow the first-order kinetics. The constant k as was calculated (as described above) to be 0.008 min^−1^, 0.013 min^−1^ and 0.028 min^−1^ for 200 ppm, 100 ppm and 20 ppm of APAP, respectively.

To verify the use of the 3D-printed TiO_2_/PS nanocomposite photocatalysts for practical environmental applications, each sample was recovered, and their efficiency tested for at least three runs. Figure 6 depicts the re-use of one of the 40% *w/w* 3D-printed TiO_2_/PS nanocomposite for three runs, against the degradation of 100 ppm APAP aqueous solution.

As one can see from Figure 6, the 3D-printed TiO_2_/PS nanocomposite photocatalysts can be successfully used at least 3 times for the photodegradation of APAP, reaching an efficiency of ~60% at the end of the 3rd run.

## 4. Summary and Conclusions

This work provides a novel experimental study concerning the successful use of TiO_2_/PS nanocomposite polymeric filaments based on 100% recycled solid polystyrene everyday products, enriched with TiO_2_ nanoparticles with mass concentrations up to 40%*w/w*, for the production of 3D photocatalytic structures/devices using a typical FDM-type 3D printer. The 3D-printed TiO_2_/PS nanocomposites were successfully used as photocatalysts for the APAP degradation. It should be noted that this is the first report of 3D-printed photocatalytic devices made of fully recycled raw materials, and with a TiO_2_ loading as high as high as 40% *w/w*.

The 3D-printed TiO_2_/PS nanocomposite samples provide promising photocatalytic properties, reaching an efficiency of almost 60% after three cycles of reuse in 200 ppm of APAP aqueous solution under UV-A irradiation, offering a novel low-cost alternate way for fabricating large-scale photocatalysts, suitable for practical applications.

## Figures and Tables

**Figure 1 nanomaterials-10-02144-f001:**
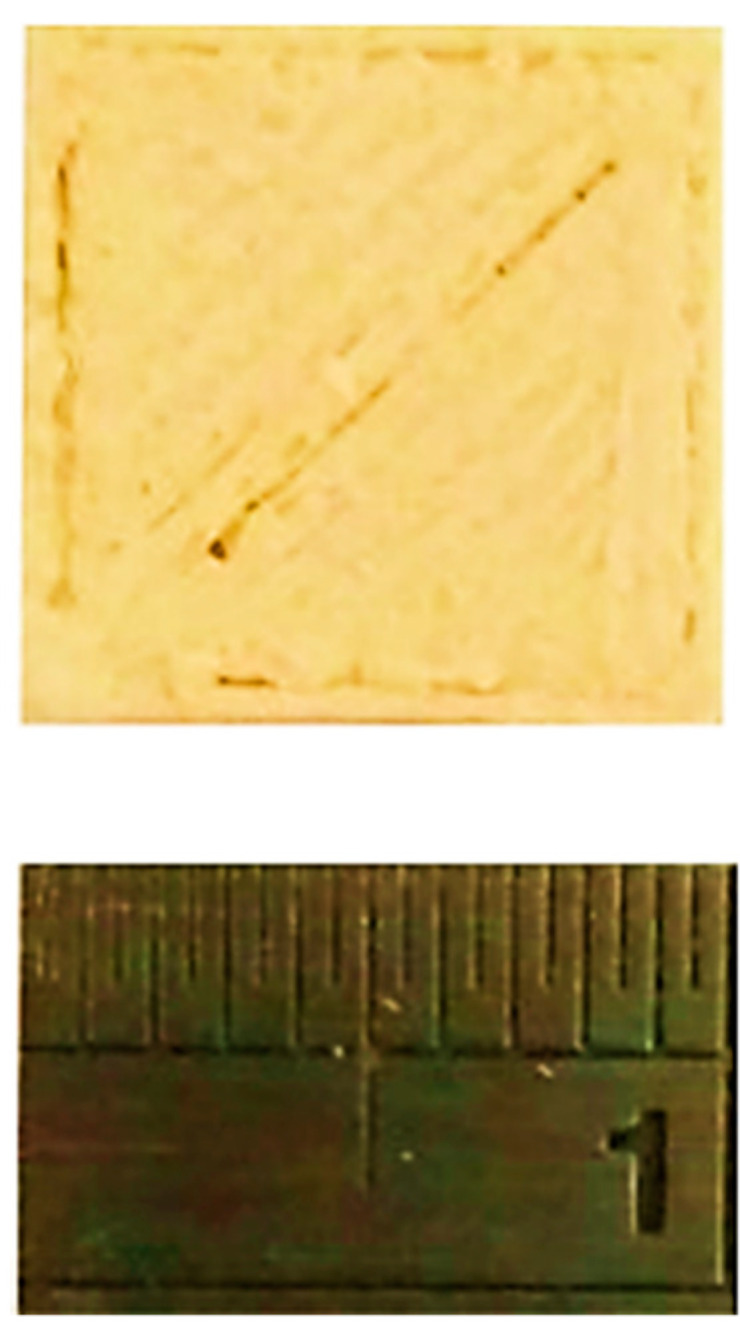
Typical photograph of a 3D-printed nanocomposite photocatalytic sample with 40% *w/w* TiO_2_ in polystyrene (PS).

**Figure 2 nanomaterials-10-02144-f002:**
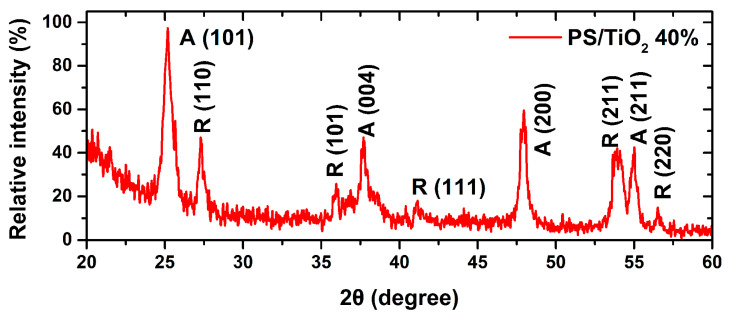
Typical X-ray diffraction (XRD) patterns for PS/TiO_2_ 3D-printed nanocomposite structures.

**Figure 3 nanomaterials-10-02144-f003:**
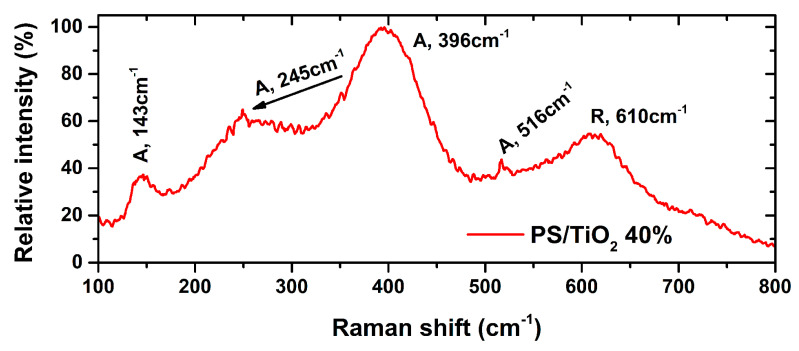
Typical Raman spectra for PS/TiO_2_ 3D-printed nanocomposite structures.

**Figure 4 nanomaterials-10-02144-f004:**
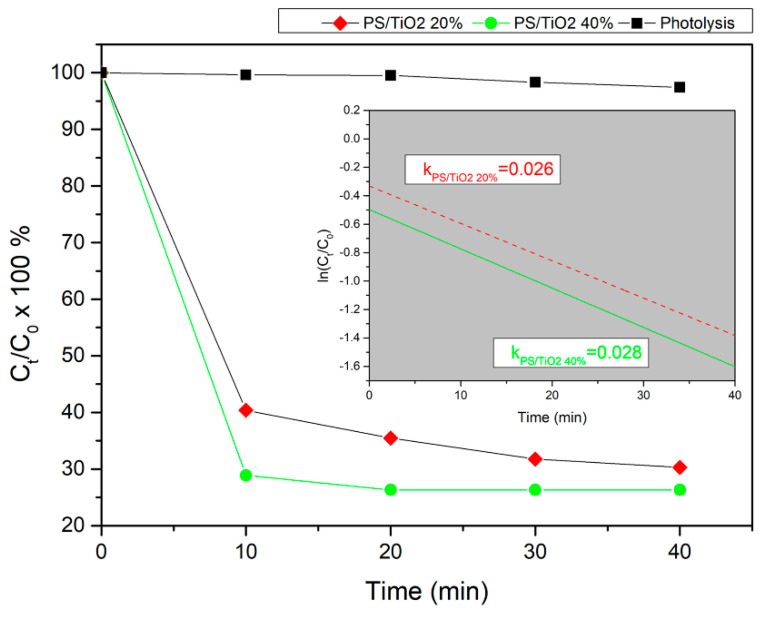
Percentage (%) acetaminophen (APAP) degradation using 20% *w/w* and 40% *w/w* [red solid rhombuses and green solid circles] TiO_2_-based 3D-printed nanocomposites under ultraviolet (UV-A) irradiation, vs. irradiation time, respectively. For comparison reasons, the photolysis curve (black solid squares) is also presented. In the inset one can see the apparent rate constants (k) of APAP degradation using 20% *w/w* and 40% *w/w* 3D printed TiO_2_/PS nanocomposite photocatalysts.

**Figure 5 nanomaterials-10-02144-f005:**
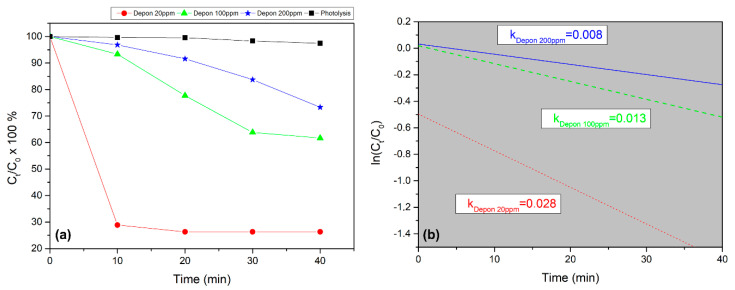
(**a**) % APAP degradation using 40% *w/w* 3D-printed TiO_2_/PS nanocomposites under UV-A irradiation, vs. irradiation time. Three different concentrations of APAP are presented; 20 ppm, 100 ppm and 200 ppm (red solid circles, green solid triangles and blue solid stars, respectively). (**b**) The apparent rate constants (k) of APAP degradation (20 ppm, 100 ppm, and 200 ppm, respectively), using 40% *w/w* 3D printed TiO_2_/PS nanocomposite photocatalysts.

**Figure 6 nanomaterials-10-02144-f006:**
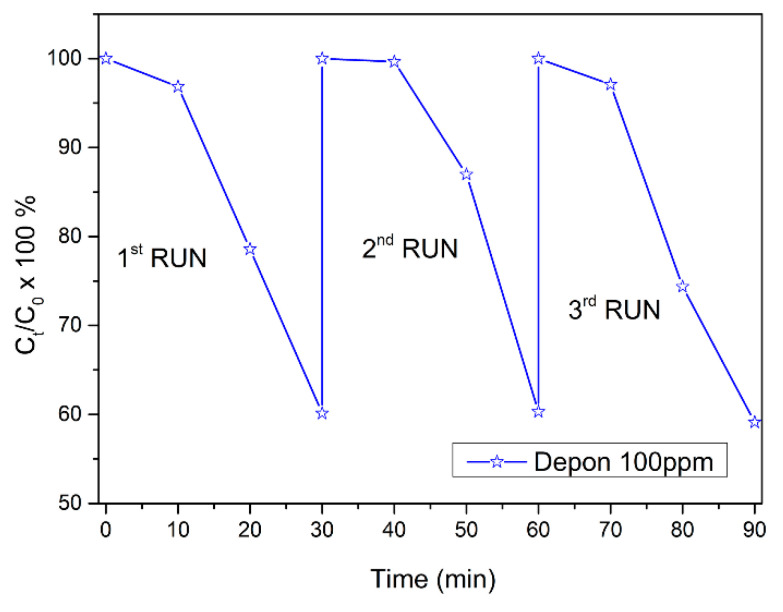
% APAP (100 ppm) degradation using a 40% *w/w* TiO_2_/PS 3D-printed nanocomposite sample under UV-A irradiation, for 3 runs of 30 min irradiation each.

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
