# Peer review of "3D Printed Fully Recycled TiO2-Polystyrene Nanocomposite Photocatalysts for Use against Drug Residues"

_nanomaterials, 2020, doi:10.3390/nano10112144_

Round 1
Reviewer 1 Report
The manuscript reports on the possibility to remove drugs as paracetamol from wastewater by using a 3D structure made by 100% recycled solid polystyrene from every day products, enriched with TiO2 nanoparticles. The paper is surely interesting and the obtained results are valid but an improvement of different sections should be provided. The manuscript is easy to follow but a grammar check could be beneficial. The scientific plan is correct in my opinion and for this reason I consider the paper suitable for publication, but before this, major revisions are needed. Anyway, I think that the manuscript does not fit completely with the aim and the scope of Nanomaterials MDPI Journal (the nanomaterial is used only to enrich 100% recycled polystyrene filament), I think that the proper collocation of the manuscript is in a Journal more focused on polymers or engineering processes.
The following improvements to the manuscript are suggested:
- The entire manuscript needs a linguistic revision (check verbs, plurals and typos).
- Fused Filament Fabrication (FFF) is listed as keyword but it is never described in the main text.
- Remove from the introduction the last paragraph, conclusions should not be anticipated into this section.
- The word Paracetamol and the acronym APAP are both used to indicate the target drug, please select only one way to refer to the target molecule and avoid such a mix in the main text.
- In the introduction change the word “alternate” with a proper one.
- In the aim of the work specify better the concentration of TiO2 in the photocatalysts object of the work.
- How do the authors check the amount of TiO2 loaded into the filament or into the 3D structure?
- Did the authors perform leaching test to probe the possible leaching of TiO2 or of other metals present as contaminants (the PS comes from recycled goods)?
- Why the authors chose a rectangular shaped structure?
- The authors collocated the 3D structure on the bottom of the quartz cell, did they test also a vertical collocation of the structure? Could the position of the structure have an influence on the remediation process?
- How many structures with the same TiO2 load have been produced and tested? Which is the reproducibility of the structure manufacturing?
- Which is the reproducibility of the photocatalytical tests with the same 3D structure? How many repeated runs have been performed with the same structure (in the text there is a little bit of confusion since the Figure 6 reports 3 runs while in the main text it has been reported 5 runs)? Please add more details in the experimental section about these issues.
- Which is the distribution of TiO2 into the filament and thus into the 3D structure? I suggest to provide EDX maps to show it.
- Did the authors make tests on the 3D structure to probe its mechanical resistance to water flux?
- Page 5: the description of the fitting results should be collocated after the description of the photocatalysis results.
Author Response
Please find bellow the answers to all R1 comments:
- The entire manuscript needs a linguistic revision (check verbs, plurals and typos).
We thank Reviewer #1 for his/her suggestion. We have checked our manuscript and made linguistic improvements.
- Fused Filament Fabrication (FFF) is listed as keyword but it is never described in the main text.
Typically, FFF (Fused Filament Fabrication) and FDM (Fused Deposition Modeling) are identical terms, however FDM is a trademarked term, while FFF is not. The processes involved in creating a 3D print for FFF and FDM is exactly the same. This is why we have added both the keywords “Fused Deposition Modeling (FDM)”and “Fused Filament Fabrication (FFF)” to the keyword section.
Remove from the introduction the last paragraph, conclusions should not be anticipated into this section.
We thank Reviewer #1 for his/her suggestion. We have changed the introduction section accordingly.
- The word Paracetamol and the acronym APAP are both used to indicate the target drug, please select only one way to refer to the target molecule and avoid such a mix in the main text.
We thank Reviewer #1 for his/her suggestion. We have changed our manuscript accordingly.
- In the introduction, change the word “alternate” with a proper one.
We have revised our manuscript accordingly.
- In the aim of the work specify better the concentration of TiO2 in the photocatalysts object of the work.
We kindly ask Reviewer #1 to check our revised manuscript, along with our comments to the other Reviewers.
- How do the authors check the amount of TiO2 loaded into the filament or into the 3D structure?
We used a type of thermogravimetric method; We have weighted a small piece of each sample on a quartz substrate, then we have heated it at ~900°C, in order to burn all organics and polymeric residuals, and finally we have weighted it again. Since TiO2 is not affected at all at such temperatures, the remaining mass is the TiO2 loading. This way we have checked that the nominal TiO2 loadings are indeed 20% and 40% w/w ± 0.5-1.0%.
We have revised our manuscript in order to make it clear to the readers.
- Did the authors perform leaching test to probe the possible leaching of TiO2 or of other metals present as contaminants (the PS comes from recycled goods)?
We really thank the reviewer for his/her comment. Although leaching of TiO2 (or other ingredients) of the 3D printed samples is indeed critical, we have performed such test. In this work, we wanted to stress the fabrication of recycled (the PS comes from recycled goods) TiO2-polystyirene nanocomposite photocatalysts for use against drug residues. Indeed, we provide evidence that we can fabricate environmentally friendly photocatalytic devices with promising photocatalytic properties, reaching an efficiency of almost 60% after three cycles of reuse in 200ppm paracetamol aqueous solutions, under UV irradiation. We shall keep in mind this comment and study the leaching of TiO2 in future work.
- Why the authors chose a rectangular shaped structure?
We have chosen the flat rectangular shaped structure in order to show that even the simplest flat 3D surfaces give high photocatalytic efficiencies. Indeed, several other structures (with higher surface-to-volume ratio) could give better photocatalytic results. On the other hand, more sophisticated structures, such as pillars, pyramids etc., would need more printing time, and since we are working with home-made nanocomposite structures the quality of the surface and the desired geometry would not be the ideal one. We kindly ask reviewer #1 to check “Nanomaterials 9 (2019) 1056” (doi: 10.3390/nano9071056), a former work of our research team, to check the surface of such 3d printed structures.
- The authors collocated the 3D structure on the bottom of the quartz cell, did they test also a vertical collocation of the structure? Could the position of the structure have an influence on the remediation process?
We thank Reviewer #1 for his/her suggestion. We have used a vertical collocation of the structure; we have not collocated the 3D structure on the bottom of the quartz cell. We have revised our manuscript in order to make it clearer.
Nevertheless, since we are talking about APAP solutions and stable TiO2/PS solid photocatalyst, the position of the structure would not have an influence on the remediation process. If we used TiO2 suspensions, the remediation would be different due to precipitation of the catalysts.
- How many structures with the same TiO2 load have been produced and tested? Which is the reproducibility of the structure manufacturing?
As already stated in our manuscript, the photocatalytic activity tests were carried out for at least five times on the 3D printed TiO2/PS nanocomposite samples to examine their stability under UV illumination, demonstrating no changes in the photocatalytic activity after five runs. Moreover, at least three structures with the same TiO2 load have been produced and tested, in order to check the reproducibility of the structure manufacturing.
We have revised our manuscript accordingly.
- Which is the reproducibility of the photocatalytical tests with the same 3D structure? How many repeated runs have been performed with the same structure (in the text there is a little bit of confusion since the Figure 6 reports 3 runs while in the main text it has been reported 5 runs)? Please add more details in the experimental section about these issues.
We state that we have performed at least 5 repeated runs, although we present three of them. Nevertheless, in order to make it clearer we have revised our manuscript, mentioning that we have performed just three of them.
- Which is the distribution of TiO2 into the filament and thus into the 3D structure? I suggest to provide EDX maps to show it.
We have tested the procedure that Reviewer #1 suggests. In the case of micrometer sized particles this is indeed valid, and we have used in order to check the distribution of the μm loading. For the case of 20-25 nm sized TiO2 particles the polymeric samples are overheated, creating volcano effect and the results of EDX are not trustworthy.
- Did the authors make tests on the 3D structure to probe its mechanical resistance to water flux?
We have not performed such tests. Since we show that the fabricated 3D printed TiO2/PS nanocomposite photocatalysts can be successfully used for at least 3 times against the photodegradation of APAP, reaching an efficiency of ~60% at the end of the 3rd run (the same like the 1st run), we feel that this kind of tests would not give something new. If we had any type of corrosion of the TiO2/PS nanocomposite photocatalysts, or any type of TiO2 immigration in water, the repeatability tests would reveal it; it that case the test the Reviewer #1 is mentioning would be necessary.
- Page 5: the description of the fitting results should be collocated after the description of the photocatalysis results.
We have collocated the description of the fitting results after the description of the photocatalysis results, according to the reviewers comment.

Reviewer 2 Report
In this paper, the authors reported the preparation of TiO2-polystyrene nanocomposite photocatalysts to eliminate drug residues in water using recycled solid polystyrene (PS) and TiO2 nanoparticles.
The flat, rectangler-shaped TiO2-polystyrene nanocomposite photocatalysts are prepared by fused deposition modeling-type 3D printer for practical applications.
The reported results in this paper are interesting and may be useful for practical applications of TiO2 phtocatalysts.
However, there are some comments as follows.
Comments
- It seem that there are some difference in photocatalytic activity of 40%w/w 3D printed TiO2/PS nanocoposite for 200ppm acetaminophen aqueous solution between Fig. 5(b) and Fig. 6.
Are there any differences in the condition such as temperature and samples between two cases?
- Fig. 1
I would like to know the detail of TiO2 nanoparticls dispersion condition at surface of 3D printed nanocomposite photocatalytic sample.
Minor comments
- Title
Polystirene → Polystyrene
- page 3, last paragraph
Raman → Raman spectra
- page 4, last line
TiO2 → TiO2
- Fig. 4, Fig. 5(b)
K → k
- figure caption
(b) → (b)
Author Response
Please find bellow th answers to all R2 comments:
- It seem that there are some difference in photocatalytic activity of 40%w/w 3D printed TiO2/PS nanocomposite for 200ppm acetaminophen aqueous solution between Fig. 5(b) and Fig. 6.
Are there any differences in the condition such as temperature and samples between two cases?
We thank Reviewer #2 for his/her comment. He/she is absolutely right. We had used a wrong legend in our plot, using 200ppm instead of the correct 100ppm APAP aqueous solution. We have changed the legend and the plot axes, and revised our manuscript. We apologize for the inconvenience.
- 1
I would like to know the detail of TiO2 nanoparticls dispersion condition at surface of 3D printed nanocomposite photocatalytic sample.
As stated in our manuscript, commercially available TiO2 nanoparticles (TiO2 P25 with a mean particle size of ~25nm, obtained from Evonic Industries AG, City, Germany) were introduced in PS/toluene solution, in order to obtain TiO2 homogeneous suspensions with 20% w/w and 40% w/w concentration in PS, respectively. PS/TiO2 was precipitated, dried, extruded and 3D printed. The nanoparticles’ concentration was 20% and 40% w/w respectively in PS in the whole mass of the 3D printed samples, not only their surface. We have revised our manuscript accordingly, in order to make it clearer for the reader.
Minor comments
- Title
Polystirene → Polystyrene
We thank Reviewer #2 for his/her suggestion. We have changed the title of our manuscript accordingly.
- page 3, last paragraph
Raman → Raman spectra
We thank Reviewer #2 for his/her suggestion. We have revised our manuscript accordingly.
- page 4, last line
TiO2 → TiO2
We thank Reviewer #2 for his/her suggestion. We have changed TiO2 to TiO2.
- 4, Fig. 5(b)
K → k
We thank Reviewer #2 for his/her suggestion. We have revised Figs. 4 and 5 accordingly.
- figure caption
(b) → (b)
We thank Reviewer #2 for his/her suggestion. We have revised our manuscript accordingly.

Reviewer 3 Report
In the current work, Authors have demonstrated an interesting approach to removal of drug residues using nanocomposite photocatalysts. From a substantive point of view, the presented work including a kinetic calculations, makes a valuable contribution to the research field.
However, from the reviewer`s duty I am forced to point out some editorial mistakes found in the text:
1) Inconsistency in using / not using the space between the numerical value and the unit (mL, oC, nm etc.).
2) Typing error in the title of the article (‘polystirene’).
3) Inconsistency in using / not using journal abbreviations in References. Entry 6 (Water Res.), entry 7 (J. Lumin.), entry 8 (Mater. Res. Bull.), entry 9 (J. Photochem. Photobiol. A), entry 12 (Dyes Pigm.), entry 27 (Water Res.), entry 46 (Chron. Pharm. Sci.), entry 47 (Environ. Pollut.) and entry 48 (Environ. Sci. Pollut. Res.).
After considering the aforementioned corrections I would recommend the publication of the present paper in Nanomaterials.
Author Response
We thank Reviewer #3 for his/her observations. We have revised our manuscript accordingly and accommodated all suggested minor corrections.
1) Inconsistency in using / not using the space between the numerical value and the unit (mL, oC, nm etc.).
The space between numerical values and the unit was removed so that now the format is consistent in the whole manuscript.
2) Typing error in the title of the article (‘polystirene’).
The error was corrected.
3) Inconsistency in using / not using journal abbreviations in References. Entry 6 (Water Res.), entry 7 (J. Lumin.), entry 8 (Mater. Res. Bull.), entry 9 (J. Photochem. Photobiol. A), entry 12 (Dyes Pigm.), entry 27 (Water Res.), entry 46 (Chron. Pharm. Sci.), entry 47 (Environ. Pollut.) and entry 48 (Environ. Sci. Pollut. Res.).
The respective inconsistency is fixed. We corrected the journal abbreviations.

Round 2
Reviewer 1 Report
The authors addressed most of the concerns I underlined during the first round of revision, so now in this form the manuscript is suitable for publication to me. Even though, as just last recommendation, I suggest to use the word degradation instead of decolorization within the main text.
Author Response
Thank you for your recommendation and help to improve our contribution. We accept the suggestion and used degradation instead of decolorization within the main text in the revised manuscript version.
